# mPLUG-PaperOwl: Scientific Diagram Analysis with the Multimodal Large Language Model

Submission ID: 3283

## ABSTRACT

Recently, the strong text understanding and generation abilities of Large Language Models (LLMs) have given rise to many tools for assisting paper reading or even writing. However, the weak diagram analysis abilities of LLMs or Multimodal LLMs greatly limit their application scenarios, especially for scientific academic paper writing. In this work, towards a more versatile copilot for academic paper writing, we mainly focus on strengthening the multi-modal diagram analysis ability of Multimodal LLMs. By parsing Latex source files of academic papers, we carefully build a multi-modal diagram understanding dataset M-Paper. By aligning diagrams in the paper with related paragraphs, we construct professional diagram analysis samples for training and evaluation. M-Paper is the first dataset to support joint comprehension of multiple scientific diagrams, including figures and tables in the format of images or Latex codes. Besides, to better align the copilot with the user's intention, we introduce the 'outline' as the control signal, which could be directly given by the user or revised based on auto-generated ones. Comprehensive experiments with a state-of-the-art Multimodal LLM demonstrate that training on our dataset shows stronger scientific diagram understanding performance, including diagram captioning, diagram analysis, and outline recommendation. The dataset, code, and model will be publicly available.

## KEYWORDS

Multimoal Large Language Model, Scientific Diagram Analysis

### ACM Reference Format:

Anonymous Authors, Submission ID: 3283. 2018. mPLUG-PaperOwl: Scientific Diagram Analysis with the Multimodal Large Language Model. In *Proceedings of Make sure to enter the correct conference title from your rights confirmation emai (Conference acronym 'XX).* ACM, New York, NY, USA, 10 pages. https://doi.org/XXXXXXX.XXXXXXX

## 1 INTRODUCTION

The strong text creation ability of the Large Language Model (LLM) [7, 39, 40, 43] inspires the development of paper-writing copilot recently, such as jenni[1]. However, existing LLMs or Multimodal LLMs are still not fully competent to assist academic paper writing due to their weak scientific diagram analysis abilities.

[1]https://jenni.ai/

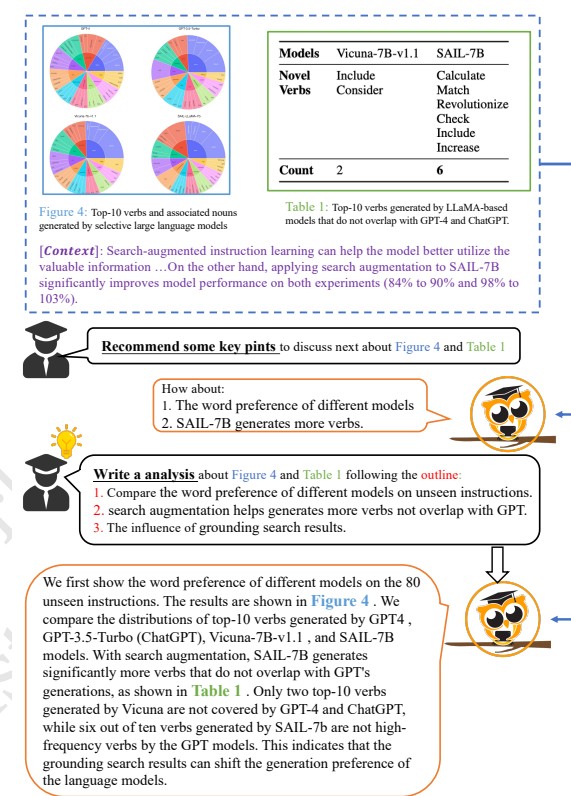

Figure 4: Top-10 verbs and associated nouns generated by selective large language models

Table 1: Top-10 verbs generated by LLaMA-based models that do not overlap with GPT-4 and ChatGPT.

| Models | Vicuna-7B-v1.1 | SAIL-7B |
|---|---|---|
| **Novel Verbs** | Include Consider | Calculate Match Revolutionize Check Include Increase |
| **Count** | 2 | 6 |

**[Context]:** Search-augmented instruction learning can help the model better utilize the valuable information …On the other hand, applying search augmentation to SAIL-7B significantly improves model performance on both experiments (84% to 90% and 98% to 103%).

**Recommend some key pints** to discuss next about Figure 4 and Table 1

How about:
1. The word preference of different models
2. SAIL-7B generates more verbs.

**Write a analysis** about Figure 4 and Table 1 following the outline:
1. Compare the word preference of different models on unseen instructions.
2. search augmentation helps generates more verbs not overlap with GPT.
3. The influence of grounding search results.

We first show the word preference of different models on the 80 unseen instructions. The results are shown in **Figure 4** . We compare the distributions of top-10 verbs generated by GPT4 , GPT-3.5-Turbo (ChatGPT), Vicuna-7B-v1.1 , and SAIL-7B models. With search augmentation, SAIL-7B generates significantly more verbs that do not overlap with GPT's generations, as shown in **Table 1** . Only two top-10 verbs generated by Vicuna are not covered by GPT-4 and ChatGPT, while six out of ten verbs generated by SAIL-7b are not high-frequency verbs by the GPT models. This indicates that the grounding search results can shift the generation preference of the language models.

**Figure 1: An illustration of paper-writing copilot for scientific diagram analysis with multiple diagrams, context, and user-revised outlines as inputs.**

As shown in Fig. 1, to assist the user in writing academic analysis about scientific diagrams, the copilot should be equipped with major three abilities. **First and most basically**, the model should be able to understand multiple diagrams of various types (figures, tables, etc.) and in different formats (image or latex). **Second**, to ensure the coherence of thesis writing, the diagram analysis should remain consistent with the preceding texts and therefore ask to model to correlate multimodal context and diagram information. **Third**, for better aligning the user's intention, the copilot should be controllable and interactable with the user. Recently, there have been many Multimodal Large Language Models (MLLMs) [2, 5, 10, 23, 24, 44, 51, 54] proposed by connecting a vision encoder with a Large Language Model as the language decoder. These MLLMs are good at chatting about a general image but poor at understanding diagrams. Some work [13, 14, 16, 25, 30, 48, 49] tried to develop MLLMs for Multimodal Document Understanding, covering tables, charts, webpages,.etc. However, these models mainly focus on strengthening

**Table 1: Comparison of M-Paper and existing Chart Understanding datasets. 'D' refers to 'Diagram'.**

| Dataset | Diagram | | | | Task | | |
| | Type | Data | Image | Avg.Num | Name | Input | Output (Avg.Token) |
|---|---|---|---|---|---|---|---|
| FigureQA [19] | Chart | Synthetic | Synthetic | 1 | VQA | D + Question | Answer (2) |
| DVQA [18] | Chart | Synthetic | Synthetic | 1 | VQA | D + Question | Answer (2) |
| PlotQA [31] | Chart | Real-world | Synthetic | 1 | VQA | D + Question | Answer (6) |
| ChartQA [27] | Chart | Real-world | Real-world | 1 | VQA | D + Question | Answer (5) |
| SciGraphQA [21] | Scientific Chart | arXiv | arXiv | 1 | VQA | D + Question | Answer (76) |
| Chart-to-Text [20] | Chart | Real-world | Real-world | 1 | Captioning | D | Caption (69) |
| VisText [38] | Chart | Real-world | Synthetic | 1 | Captioning | D | Caption (63) |
| SciCap [15] | Scientific Chart | arXiv | arXiv | 1 | Captioning | D | Caption (20) |
| SciCap+ [47] | Scientific Chart | arXiv | arXiv | 1 | Captioning | D+Mentioned Paragraph+OCR | Caption (31) |
| M-Paper | Scientific Figure&Table | arXiv | arXiv | 1.3 | Captioning | D+Preceding Texts | Caption (58) |
| | | | | | Outlining | D+Preceding Texts | Outline (36) |
| | | | | | Analysis | D+Preceding Texts+Outline | Analysis (135) |

the vision comprehension of a single diagram and can't generate detailed scientific analysis.

In this work, to develop scientific diagram analysis skills for the paper-writing copilot, we first build a comprehensive dataset **M-Paper** to support the learning of the three critical abilities mentioned above. By parsing Latex source files of academic papers, we carefully extract diagrams in both image and latex formats and align them with their captions and paragraph analysis. To simulate two main scenarios of scientific diagram understanding, we design two main tasks, namely **Multimodal Diagram Captioning** and **Multimodal Diagram Analysis**, which aim to generate concise captions and detailed analysis for multiple diagrams, respectively. Besides diagrams, we also provide preceding texts of the thesis, namely [*Context*], as inputs to teach the model how to utilize background knowledge and maintain fluency with previous content. Furthermore, to better align users' writing intentions, we design [*Outline*] as control signals, which are comprised of concise key points to be covered in the analysis. We utilize the ChatGPT to construct [*Outline*] based on ground-truth paragraph analysis and feed it as another input for *Multimodal Diagram Analysis*. For more user-friendly interaction, automatically recommending [*Outline*] by the copilot could inspire users or reduce interaction costs. Thus, we set up another **Outline Recommendation** task to make the copilot more versatile and user-friendly. For accurately evaluating the diagram analysis quality, besides classical captioning metrics (e.g. CIDEr [42]) based on n-gram matching, we carefully designed a CIDEr$^{gpt}$ score to measure both n-gram and semantic similarity with the help of ChatGPT.

We benchmark multiple state-of-the-art MLLMs on our dataset, validating the challenge of our three tasks. Based on the DocOwl [48], we perform instruction-tuning on a combination of training data from three tasks and propose a strong generalist as the baseline, named PaperOwl. Comprehensive experiments validate the effectiveness of introducing [*Context*] and [*Outline*] as inputs. We further perform ablation studies about vision encoding to provide insights about model improvement, such as increasing the image resolution and enhancing the ability to correlate multiple diagrams.

In summary, our contributions are three-fold:

- We build the first high-quality scientific diagram analysis dataset M-Paper to support the learning of correlating multiple diagrams, keeping consistency with the preceding content, and being interactable with users.
- Simulating Real-world paper-writing scenarios, we carefully design three multimodal tasks and propose a GPT-based metric, CIDEr$^{gpt}$, to measure the analysis quality by considering both detailed n-gram and overall semantic similarity.
- We carefully tune a generalist based on an existing MLLM and perform comprehensive experiments to validate the effectiveness of multimodal inputs and training strategies.

## 2 RELATED WORK

**Text-only Paper Understanding** [1, 3, 4, 26, 34, 35] focuses on text and citation graph comprehension in academic papers. Such models are competent for a number of text-only thesis comprehension tasks, including information extraction, text classification, paper summarization, or citation recommendation. Benefiting from the strong text understanding ability of Large Language Models (LLMs), many LLM-based tools have been developed as paper-reading assistants, such as ChatDoc[2], ChatPDF[3] and Zhiwen[4]. However, they are still not capable of assisting paper writing due to a lack of multimodal abilities to understand vision information and generate comprehensive diagram analyses, which are indispensable in scientific papers.

**Multimodal Document Understanding** aims to develop multimodal comprehension abilities for images with rich text information, including charts [20, 27, 38, 46], tables [9, 33], documents [29, 36, 37, 52] and infographic images [28], etc. Task formats of these work range from Information Extraction [36, 37], Question Answering [27–29], Natural Language Inference [9] to Image Captioning [15, 20, 38, 47]. Datasets about Chart Understanding [15, 18–21, 27, 31, 38, 47] are most relevant with our dataset M-Paper. Major differences with these works are shown in Table 1. Compared with

---

[2]https://www.chatdoc.com/
[3]https://www.chatpdf.com/
[4]https://tongyi.aliyun.com/zhiwen

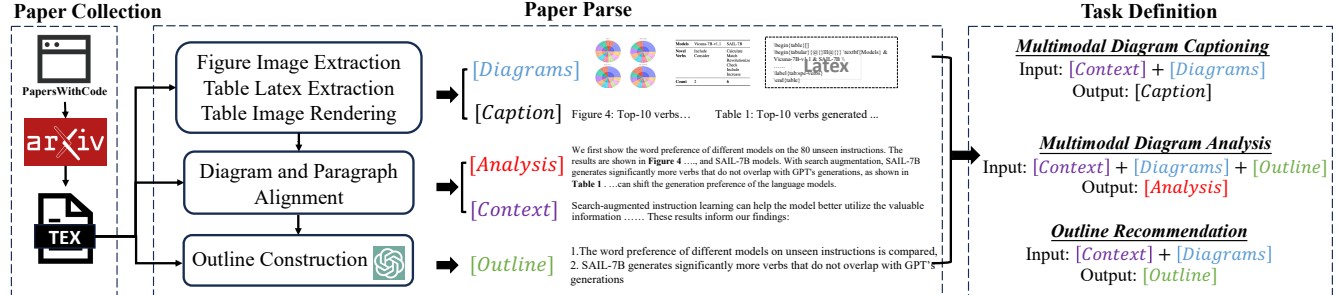

**Figure 2: The pipeline of M-Paper construction and definition of our three tasks.**

existing datasets, whether VQA or Chart Captioning, our Multimodal Diagram Analysis task provides much longer texts as targets, which enable models to learn more comprehensive analysis about diagrams. The most similar work to ours is the SciCap+ [47], which aims to generate captions of scientific charts and provides the chart, OCR results, and the first paragraph mentioning the chart as inputs. There are three major differences between our Diagram Analysis task and SciCap+. **First**, the understanding object in our task can be multiple diagrams, even a combination of charts and tables, while SciCap+ just inputs one chart. **Second**, the output of our Diagram Analysis task is a detailed paragraph analysis (average of 135 tokens) rather than a concise caption (average of 31 tokens). **Finally**, SciCap+ focuses on providing more accurate captions for scientific charts with the help of the body of the paper and OCR information. Our work aims to build a paper-writing copilot and therefore provides preceding texts as context to keep writing coherence and outlines as control signals to follow users' intentions.

**Multimodal Large Language Models** Recently, some works [5, 12–14, 16, 25, 45, 48, 49, 53] have proposed Multimodal Large Language Models with visually-situated text understanding ability. For example, UReader [49] performs instruction tuning on an ensembled dataset covering various types of images and designs a Shape-adaptive Cropping Module to process high-resolution document images. However, these MLMMs are still far from acting as a paper-writing copilot for scientific diagram analysis due to main two shortages. First, they can only generate a short answer or description and lack comprehensive diagram analysis abilities. Second, they are all trained to understand a single image, and thus can't correlate context and multiple figures or tables for accurate multimodal analysis. To empower MLMMs with such abilities, we carefully build a scientific diagram analysis dataset M-Paper based on academic papers. Fineunted on this dataset, our PaperOwl shows stronger multimodal diagram analysis abilities and moves a step closer to paper-writing copilot.

## 3 M-PAPER

Towards developing a paper-writing copilot, this work first builds a dataset M-Paper to empower models with abilities of multimodal scientific diagram captioning and analysis. The construction pipeline and task definition of M-Paper are shown in Fig. 2.

### 3.1 Paper Collection

The arXiv[5] is an open-access repository of electronic preprints and postprints, consisting of scientific papers in computer science, mathematics, physics, etc. Due to the field gap, diagrams, writing, and analysis styles are quite different across these fields. In this work, we chose 'Computer Science' as the study object. Due to that not all papers are reviewed by peers before posting, the paper quality in arXiv varies a lot and low-quality papers may hurt the model's logical analysis abilities. Considering PapersWithCode[6] is a community-driven platform for learning about state-of-the-art research papers on machine learning, the quality of papers listed in PapersWithCode is relatively more reliable. Therefore, with the PapersWithCode API[7], we collect 48k arXiv ids, ranging from 2012 to 2023, covering 15 categories and then download their corresponding Latex source files following official instructions[8].

### 3.2 Paper Parsing

PDF and Latex are two kinds of commonly used file formats in paper-related research. In this work, we choose to parse Latex source files for two main reasons. Firstly, by comparing the content in the `\ref{.}` tag and `\label{.}` tag in Latex files, it's easy to accurately correlate diagrams with paragraph analysis in the body of papers. Secondly, the Latex syntax is a more natural and general format for LLM to understand or generate diverse texts, including plain text and mathematical expression, etc. Taking into account these two points, Latex-style text understanding and generation is more suitable for a paper-writing copilot. Following S2ORC [26], we first parse Latex source files into XML format and then extract diagrams and correlate them with captions and paragraphs.

**Text Cleaning.** Towards paper-writing copilot, this work focuses on improving the model's multimodal diagram analysis abilities and pays little attention to other writing abilities, such as equation generation or citation recommendation. It's virtually impossible to infer both formulas and paper references from diagrams or preceding texts. Therefore, we further clean paragraph texts by filtering such unnecessary information. Concretely, we first replace all citation tags `\cite{.}` with a special token '<cite>' to remove

---

[5]https://arxiv.org/
[6]https://paperswithcode.com/sota
[7]https://paperswithcode-client.readthedocs.io/
[8]https://info.arxiv.org/help/api/basics.html

citation reference. To avoid generating too-long equations, paragraphs containing equations with > 40 chars are dropped.

**Table Image Rendering.** Both figures and tables are widely used in scientific academic papers. By parsing the Latext source file, it's easy to align figure reference with figures in image format (e.g., 'jpg') by the '\includegraphics' tag. But for tables, there are only Latex codes and no image-format files provided. Towards wider application scenarios, a diagram analysis copilot is necessary to understand tables in both latex and image formats. To support the learning of such abilities, we further collect table images as inputs. Directly extracting table bounding boxes from PDF-format papers with pdf-parsing tools (e.g., GROBID[9]) and then cropping table image is a naive way. However, due to the diverse layout in scientific papers, table coordinates given by such tools are not accurate enough. In this work, we collect accurate table images by following three steps. Firstly, we revise the Latex source file to ensure that each table will occupy a separate page after PDF compiling. This operation could greatly reduce the difficulty of table recognition. Then, for each PDF page containing a table, we utilize the classical Edge Detection algorithm Canny [8] to recognize the table bounding box. Finally, the table image is cropped from the PDF page according to the table coordinates. It's worth noting that, to also support the table captioning task and avoid leaking caption information in the cropped table image, the content within the '\caption{.}' tag is removed before PDF compiling.

**Outline Construction.** During paper writing, for an identical figure or table, even different co-authors can give analysis from different perspectives. Therefore, although a paper-writing copilot can give a comprehensive analysis of a diagram, its analysis can still go against the author's wishes or be inconsistent with the preceding texts. To better cater to users' intentions, we propose to use the 'outline' as the intermediate control signal during diagram analysis. Besides directly generating the paragraph analysis, the copilot should also be able to analyze the diagram more accurately following provided key points, namely 'outline'. During paper writing, the outline could given by users or generated by the copilot and revised by users.

For developing such a versatile and controllable copilot, it's necessary to construct appropriate training data for outline generation and analysis generation with outlines. To construct such training samples, in this work, we utilize the GPT-3.5[10] to generate corresponding outlines for each paragraph by in-context learning. More details can be found in the supplementary material.

## 3.3 Task Definition

After processing Latex source files as mentioned above, we carefully organize these data to support the training and test of multiple tasks designed for the paper-writing copilot, including *Multimodal Diagram Captioning*, *Multimodal Diagram Analysis*, and *Outline Recommendation*.

**Multimodal Diagram Captioning.** Different from conventional Image Captioning which aims to describe the attributes and relation between objects, Diagram Captioning requires the model to accurately summarize the content in the figure or table, including

---

[9]https://github.com/kermitt2/grobid
[10]https://openai.com/blog/chatgpt

**GT:** Table 5 shows that the translation of in-domain natural inputs improve significantly after applying TST BT. We also found that TST BT still improve translation of out-of-domain natural inputs.

| Prediction | CIDEr | $F1^{gpt}$ | $CIDEr^{gpt}$ |
|---|---|---|---|
| A: We found that TST BT significantly improves the translation of in-domain natural inputs. | 13.52 | 0.66 | **8.92** |
| B: We further perform experiments to test the generalizability of TST BT. Table 5 shows experimental results | **14.11** | 0.00 | 0.00 |

**Figure 3: A case of the comparsion of CIDEr and** $CIDEr^{gpt}$.

some concrete mathematical symbols and proper nouns. Besides, due to partial diagrams being a combination of sub-diagrams, it also asks the model to correlate multiple images. Further, the table during paper-writing can be an image or Latex code, which requires the model to understand different formats of input.

By parsing the Latex source file, it's easy to get diagram captions by extracting content from the '\caption{.}' tag. For generating captioning more consistent with the paper content and better-mentioning prop nouns, we also provide preceding text as the textual input, denoted as [*Context*]. To keep the completeness of semantics, the preceding text is comprised of multiple un-truncated paragraphs before the first reference of the diagram, with max 512 tokens. Thus, the input of Multimodal Diagram Captioning is a triplet of ⟨[*Context*], [*Diagrams*], [*Inst*]⟩, where [*Diagrams*] can be diagram images or Latex code of a table, [*Inst*] is the instruction.

Following classical image captioning tasks, we utilize BELU [32], METEOR [6], ROUGE-L [22], and CIDEr [41] as evaluation metrics. The CIDEr is valued most because it puts higher weight on rarer tokens (e.g., proper nouns), which are more informative.

**Multimodal Diagram Analysis.** Much more difficult than writing a caption, Diagram Analysis requires the model to generate a paragraph analysis according to multiple diagrams, even a combination of figures and tables. Besides, diagram analysis is more open-ended than captioning. Different people can analyze a diagram from quite different perspectives. As a paper-writing copilot to improve writing efficiency, the diagram analysis should follow users' intentions as well as possible. Therefore, besides providing the preceding text like the Multimodal Diagram Captioning task to imply the author's intention, we further design the 'outline' as the explicit control signal, which instructs key points to discuss with diagrams. Overall, the input of Multimodal Diagram Analysis is a quartet of ⟨[*Context*], [*Outline*], [*Diagrams*], [*Inst*]⟩.

Captioning metrics are not quite suitable for paragraph analysis because they mainly measure the n-gram similarity and neglect overall semantic matching. To better evaluate the analysis quality, we design a metric to measure the semantic similarity based on GPT 3.5, namely $F1^{gpt}$. Concretely, given the predicted analysis and the ground-truth one, we first prompt the GPT to extract their key points in the list format, respectively. Then, we prompt GPT to judge whether each pair of predicted key points and ground-truth key points matched or not. Finally, we calculate the semantic precision, recall, and F1 score ($F1^{gpt}$) based on GPT's judgment. Detailed

**Table 2: Statistics of M-Paper.**

| Task | | Train | Val | Test |
|------|------|-------|-----|------|
| Diagram | paper | 46,649 | 479 | 455 |
| Captioning | sample | 343,546 | 1,131 | 1,133 |
| Diagram | paper | 40,567 | 412 | 449 |
| Analysis | sample | 267,476 | 1,087 | 1,195 |
| Outline | paper | 2,548 | 543 | 577 |
| Recommendation | sample | 78,041 | 3,425 | 3,442 |

prompts for these two steps can be found in the supplementary material. The $F1^{gpt}$ is good at measuring semantic similarity but hard to assess the fine-grained quality of detailed descriptions, which is rather what CIDEr is good at. For paragraph analysis, accurately describing key points is more important and we are more tolerant of the form of expression. Considering $F1^{gpt}$ reflects the percentage of mentioning key points and CIDEr measures the n-gram similarity of the whole paragraph. we therefore multiply the CIDEr with $F1^{gpt}$ as the final evaluation metric $CIDEr^{gpt}$, where $F1^{gpt}$ plays a critical role. As shown in Fig. 3, prediction A gets a lower CIDEr score because it mentions fewer n-grams within ground truth. However, it describes semantics more accurately and therefore gets a higher $CIDEr^{gpt}$ score.

**Outline Recommendation.** Towards a user-friendly paper-writing copilot, the 'outline' can be given directly by users or generated by the copilot and then revised by the user. So recommending outlines accurately is also an important ability for inspiring users or improving writing efficiency. To develop such ability, we also design an Outline Recommendation task, where the input can be $\langle[Context], [Inst]\rangle$ or $\langle[Context], [Diagrams], [Inst]\rangle$ and the target is $[Outline]$. Captioning metrics are used to evaluate this task.

Diverse instructions for these three tasks can be found in the supplementary material.

### 3.4 Statistic

**Paper Category.** M-Paper contains 48,688 papers from more than 15 categories, covering almost all popular research directions in 'Deep Learning', especially Computer Vision (CV) and Natural language Processing (NLP). The detailed category distribution can be found in the supplementary material.

**Dataset Splits.** Table 2 shows the split statistic of *Multimodal Diagram Captioning*, *Multimodal Diagram Analysis* and *Outline Recommendation*. For each task, there is no paper overlap across the training, validation and test splits. Both *Multimodal Diagram Captioning* and *Multimodal Diagram Analysis* cover more than 40k papers and provide sufficient training samples. As for *Outline Recommendation*, considering that 'outlines' are just intermediate control signals used to interact with users, we don't expect perfect quality of generated outlines. Thus only partial papers are processed to support the training and test of this task.

**Diagram.** As shown in Fig. 4, the distribution of diagram counts varies across different tasks. For *Multimodal Diagram Analysis*, there are more than 25% samples with multiple diagrams as inputs, much more than *Multimodal Diagram Captioning*. This indicates that

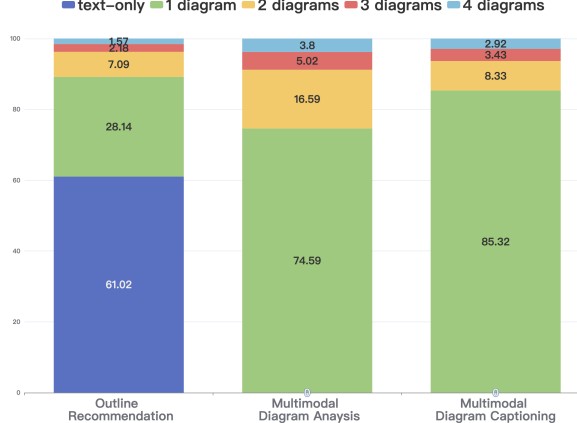

**Figure 4: The distribution (%) of diagram count across 3 tasks.**

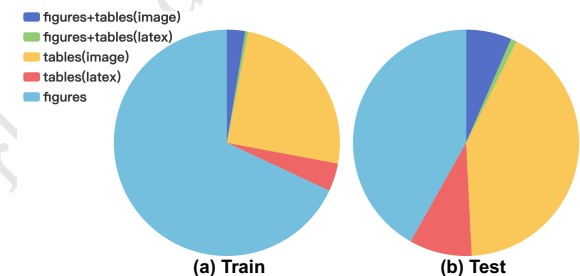

**Figure 5: The distribution of diagram types on the training and test set of Multimodal Diagram Analysis.**

**Table 3: Token statistic of different textual components.**

| | Context | Outline | Table Latex | Caption | Analysis |
|------|---------|---------|-------------|---------|----------|
| Mean | 410 | 36 | 177 | 58 | 135 |
| Max | 512 | 126 | 256 | 256 | 256 |

correlating multiple diagrams is a major challenge for *Multimodal Diagram Analysis*. Besides, Fig. 5 shows the distribution of diagram types in *Multimodal Diagram Analysis* task. Our dataset is not limited to a single diagram type but a fusion of figures and tables in the form of images or latex codes. Especially, to better evaluate analysis ability on different diagram types, we slightly balance the diagram type distribution in the test set.

**Token Length.** Table 3 presents the token length statistic of different textual components in our tasks. The average token length of the caption is much smaller than the paragraph analysis, indicating the *Multimodal Diagram Analysis* task requires a more comprehensive diagram understanding. Besides, the length of the 'outline' is far from the 'analysis', showing that the input 'outline' will not leak too much information about the target analysis but just point out some key points to discuss.

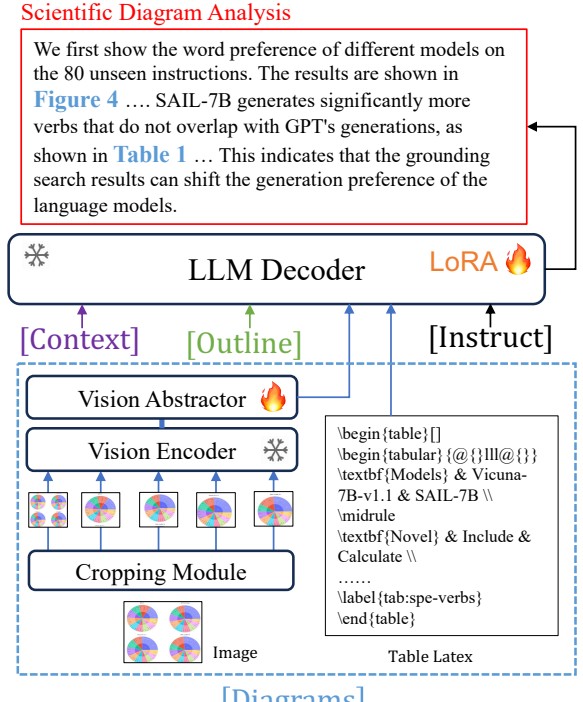

**Figure 6: The overall architecture of PaperOwl.**

## 4 MPLUG-PAPEROWL

Existing Multimodal Large Language Models (MLLMs) [5, 24, 50, 54] follow a three-module framework, consisting of a vision encoder, a vision-to-text connector, and a Large Language Model as the language decoder. Models with such a framework are easy to adapt to our multimodal tasks by constructing image-text interleaved sequences. In this work, we choose one of the state-of-the-art MLLMs: mPLUG-DocOwl [48] as the base model to perform instruction-tuning on our M-Paper.

### 4.1 Model Architecture

The overall architecture of PaperOwl is shown in Fig. 6.

**Cropping Module.** Following UReader [49], to better recognize texts in the image, we utilize a parameter-free Cropping Module to cut a 448x448 image to 4 sub-images of 224x224 resolution and then feed each sub-image to the following Vision Encoder independently.

**Vision Encoder.** The ViT-L/14 [11] is utilized as the Vision Encoder, comprised of 24 transformer layers with 16 attention heads and the dimension of hidden states set to 1024. For each image $I$ in the $[Diagrams]$, it's represented as a sequence of visual features $V = \{v_1, ..., v_n\}$ after the Vision Encoder.

**Vision Abstractor.** The Vision Abstractor is used to aggregate valuable vision semantics and align visual features to the language decoder. It consists of 6 transformer layers with 8 attention heads and the dimension of hidden states is set as 1024. With 64 learnable tokens $Q = \{q_1, ..q_k\}$ as the query, the concatenated sequence $[V : Q]$ as the key and value, the visual features are finally condensed to $\hat{V} = \{\hat{v}_1, ..., \hat{v}_k\}$ after cross attention.

**Language Decoder.** The architecture of Language Decoder is the same as LLaMA-7B [40]. To adapt to vision-and-language tasks and alleviate catastrophic forgetting, LoRA [17] is utilized in the LLM with the rank set as 8.

### 4.2 Model Training

**Data.** To develop a versatile paper-writing copilot for scientific diagram understanding, we aim to perform instruction-tuning to enhance an existing MLLM to be a generalist capable of Multimodal Diagram Captioning, Multimodal Diagram Analysis, and Outline Recommendation. Therefore, the training data is a combination of three tasks. Besides, for *Multimodal Diagram Analysis*, to avoid the model heavily relying on 'outline' to guess paragraph analysis, samples removing outlines from inputs are also added to the training data to strengthen vision understanding ability. Finally, the total number of instruction-tuning samples is 702,247.

**Details.** Following most MLLMs [24, 50, 54], the Vision Encoder in the PaperOwl is frozen during instruction-tuning to avoid hurting the strong vision representation ability learned from large-scale vision-and-language pretraining. The Vision Abstractor is fine-tuned to better learn how to filter useful visual diagram information for generating analysis. The raw parameters of LLaMA-7B are frozen, and only the LoRA in the Language Decoder is updated to learn the analysis logic of academic papers. Our model is trained for 10 epochs with the learning rate set as $1e - 4$ and the batch size as 256, costing 64 A100 days.

## 5 EXPERIMENTS

### 5.1 Comparison with SOTA MLLMs.

We first compare the zero-shot performance of existing MLLMs on our three tasks. As shown in Table 4, mPLUG-Owl [50] achieves the worst performance, showing the importance of high resolution for our tasks. After increasing image resolution, mPLUG-Owl2 [51] and LLaVA 1.5 [23] outperform the other 3 models trained with multimodal document understanding samples on *Multimodal Diagram Analysis* task. Besides, UReader [49], a model fine-tuned only on document benchmarks, achieves the worst analysis performance. This validates that existing multimodal document understanding data is far from energizing the comprehensive diagram analysis ability of MLLMs and may cause overfitting on question answering or information extraction benchmarks. However, Owl2, LLaVA 1.5, and Qwen-VL all optimize the whole LLM during instruction-tuning while UReader and DocOwl only tune the LoRA. Considering performance on three tasks and training costs, we finally chose DocOwl as our basic model. After fine-tuning with a combination of three tasks, PaperOwl achieves much better performance across three tasks, validating the effectiveness of M-Paper for developing a scientific diagram analysis copilot.

### 5.2 Ablation Study

For comprehensively analyzing critical elements for developing a scientific diagram analysis copilot, we perform sufficient comparison experiments to validate the effectiveness of $[Context]$ and $[Outline]$, and present the influence of vision encoding strategies.

**Context Influence.** For *Multimodal Diagram Captioning* and *Multimodal Diagram Analysis* tasks, we provide $[Context]$ as auxiliary

**Table 4: The performance comparison with state-of-the-art Multimodal Large Language Models on three tasks. B4, R, M, C and C$^{\text{gpt}}$ represents BLEU4, ROUGE-L, METEOR, CIDEr and CIDEr$^{\text{gpt}}$, respectively. 'underline' means the best zero-shot performance. 'Img' refers to the image resolution during training and inference. 'Doc' and 'Text' refer to using multimodal document and text-only instruction tuning data during training or not.**

| Model | Setting | | | Diagram Captioning | | | | Outline Recommendation | | | | Diagram Analysis | | | | | |
|---|---|---|---|---|---|---|---|---|---|---|---|---|---|---|---|---|---|
| | Img | Text | Doc | B4 | R | M | C | B4 | R | M | C | B4 | R | M | C | $F1^{gpt}$ | C$^{\text{gpt}}$ |
| mPLUG-Owl [50] | 224 | ✓ | ✗ | 0.36 | 8.60 | 5.30 | 0.74 | 0.62 | 9.12 | 8.55 | 0.32 | 2.48 | 15.12 | 14.67 | 0.53 | 0.21 | 0.15 |
| mPLUG-Owl2 [51] | 448 | ✓ | ✗ | 1.62 | 10.33 | 5.30 | 5.63 | 1.30 | 11.99 | **10.48** | 2.71 | 6.92 | 19.65 | 14.96 | 11.85 | 0.25 | 3.89 |
| LLaVA 1.5 [23] | 336 | ✓ | ✗ | 0.97 | 10.71 | 6.78 | 2.74 | 1.32 | 11.79 | 10.46 | 0.79 | 6.11 | 18.83 | 12.43 | 13.70 | 0.20 | 4.64 |
| Qwen-VL [5] | 448 | ✓ | ✓ | 1.84 | 7.64 | 6.61 | 2.31 | 1.32 | 7.29 | 8.52 | 0.53 | 6.72 | 10.26 | 10.74 | 3.68 | 0.27 | 1.39 |
| UReader [49] | 448 | ✗ | ✓ | 0.56 | 9.84 | 3.34 | 5.95 | 0.25 | 8.17 | 2.88 | 4.59 | 1.22 | 10.59 | 4.33 | 1.02 | 0.05 | 0.05 |
| DocOwl [48] | 448 | ✓ | ✓ | 0.87 | 10.40 | 3.64 | 8.08 | 0.45 | 9.20 | 5.98 | 2.51 | 1.90 | 14.33 | 10.28 | 4.78 | 0.19 | 1.23 |
| PaperOwl | | | | **2.37** | **18.31** | **7.19** | **25.50** | **2.16** | **17.96** | **7.33** | **30.65** | **14.74** | **29.91** | **17.38** | **22.98** | **0.40** | **11.62** |

**Table 5: The ablation study about whether utilizing [Context] during training and testing.**

| | Context | | Captioning | | | Analysis | | | |
|---|---|---|---|---|---|---|---|---|---|
| | Train | Test | R | M | C | R | M | C | C$^{\text{gpt}}$ |
| r1 | ✗ | ✗ | 15.43 | 5.45 | 14.67 | 16.56 | 8.71 | 4.45 | 1.47 |
| r2 | ✗ | ✓ | 16.62 | **6.82** | 17.72 | 14.44 | 7.66 | 2.87 | 0.94 |
| r3 | ✓ | ✓ | **17.08** | 6.76 | **21.36** | **19.25** | **10.97** | **7.02** | **1.81** |

**Table 6: The abltion study about the influence of [Outline] for Multimodal Diagram Analysis performance.**

| | Outline Usage | | B4 | R | M | C | $F1^{gpt}$ | C$^{\text{gpt}}$ |
|---|---|---|---|---|---|---|---|---|
| | Train | Test | | | | | | |
| r1 | ✗ | ✗ | 6.28 | 19.25 | 10.97 | 7.02 | 0.18 | 1.81 |
| r2 | ✗ | gpt | 7.23 | 19.86 | 11.24 | 8.99 | 0.22 | 3.10 |
| r3 | gpt | ✗ | 6.42 | 19.47 | 11.15 | 7.90 | 0.17 | 2.13 |
| r4 | gpt | auto | 5.98 | 19.58 | 11.23 | 9.10 | 0.19 | 2.59 |
| r5 | gpt | gpt | **15.27** | **30.36** | **17.49** | **21.85** | **0.41** | **11.23** |

inputs to implicitly represent users' next writing intention and provide some background information of proper nouns. We first utilize Owl [50] as the basic model to study whether using [Context] during training and testing. All models are just trained on captioning and analysis tasks and remove [Outline] from inputs. As shown in Table 5, for the model trained without [Context], providing [Context] during inference could improve the captioning performance (r2 vs r1), showing [Context] is critical for Diagram Captioning. However, adding [Context] only in testing hurts the analysis performance, indicating the model is hard to balance the comprehension of preceding texts and multiple diagrams for paragraph analysis generation. After adding [Context] in training, the model achieves better performance on both two tasks (r3 vs r2), showing that for better scientific diagram comprehension, it's necessary to incorporate [Context] during both training and inference. **Outline Influence.** To better align the diagram analysis from a paper-writing copilot with users' intention, we propose to introduce [Outline] as explicit control signals. For validating the effectiveness of [Outline], we further compare variants of Owl about whether utilizing [Outline] during training and testing. As presented in Table 6, for models trained with [Outline] as inputs or not, adding [Outline] during inference could both improve the performance (r2 vs r1, r5 vs r3), showing 'Outlines' is an effective control signal for guiding diagram analysis. Besides, even adding pseudo [Outline] generated by the model itself as inputs, the analysis quality could also be improved (r4 vs r3). This indicates that 'recommending [Outline] first and then generating diagram analysis' may be a better two-step framework, where the user could also control the copilot by slightly revising the recommended [Outline]. Finally, trained with [Outline] makes a significant improvement (r5 vs r2),

validating the necessity of learning how to correlate [Context], [Outline], and [Diagrams] for scientific diagram analysis.

**Vision Encoding Strategies.** For vision-and-language tasks, the visual features play a big role in the final performance. In this section, we compare the influence of different vision-representing strategies, including image resolution, whether to fine-tune the Vision Abstractor, and whether to crop the image. As shown in Table 7, during instruction-tuning, freezing the Vision Abstractor greatly hurt the diagram analysis performance (r1 vs r2), validating that fine-tuning the Vision Abstractor is important for adapting an existing MLLM for professional diagram understanding. Besides, at the condition of freezing the Vision Encoder, directly increasing the image resolution and expanding patch position embeddings by bicubic interpolation doesn't bring significant improvement (r3 vs r2), showing that only finetuning the Vsion Abstractor is not enough to adapt to higher-resolution images. When equipped with a parameter-free Cropping Module as UReader [49] to cut the 448x448 image to 4 sub-images of 224x224 resolutions, the model achieves significantly better performance on the diagram captioning task (r4 vs r2), showing that when the Vision Encoder is frozen, cropping images is a better solution for leveraging higher-resolution images. But, compared with the diagram captioning task, the cropping module still brings a weak improvement to the analysis task. This is mainly because the cropping module results in too many visual tokens (max 1024 tokens from 16 sub-images) and therefore greatly increases the difficulty of multimodal understanding for the language decoder. This shows that how to better encode high-resolution images and balance multimodal inputs is a major challenge for the *Multimodal Diagram Analysis* task.

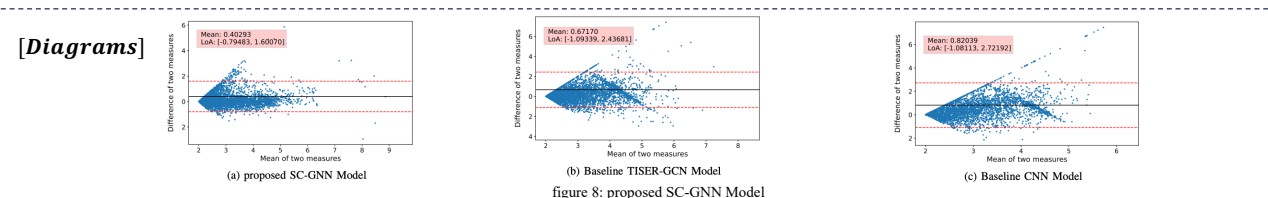

**[*Diagrams*]**

(a) proposed SC-GNN Model     (b) Baseline TISER-GCN Model     (c) Baseline CNN Model

figure 8: proposed SC-GNN Model

**[*Context*]:** For the proposed SC-GNN model (Fig. 8 ), the Bland-Altman plot shows a mean difference of 0.40 and limits of agreement of [-0.79, 1.60]......This implies that the baseline models have difficulty accurately predicting ground motion intensities for shallow earthquakes.

**GT:** In contrast, the baseline TISER-GCN model (Fig. 8 ) presents a mean difference of 0.67 and limits of agreement of [-1.09, 2.43]. The increased mean difference compared to the SC-GNN model suggests that the TISER-GCN model predictions are less accurate. Additionally, the wider limits of agreement indicate a higher level of variability in the model's performance

---

**Multimodal Diagram Analysis with GPT-extracted Outline**

**[*Outline*]:** The TISER-GCN model has less accurate predictions and higher variability than the SC-GNN model.

**[*Instruction*]:** Drawing from the preceding multimodal content and adhering to the outlined, write a thorough and coherent paragraph analysis.

**PaperOwl** : The TISER-GCN model (Fig. 8 ) shows a mean difference of 0.60 and limits of agreement of [-1.10, 2.10]. This indicates that the TISER-GCN model predictions are, on average, less accurate than the SC-GNN model predictions. The wide range of the limits of agreement suggests that the model's performance is inconsistent across the range of earthquake intensities.

---

**Multimodal Diagram Analysis without Outline**

**[*Instruction*]:** Drawing from the preceding multimodal content, write a thorough and coherent paragraph analysis.

**PaperOwl** : The proposed SC-GNN model (Fig. 8 ) shows a mean difference of 0.40 and limits of agreement of [-0.79, 1.60]. This indicates that the SC-GNN model predictions are, on average, in good agreement with the true observed intensities. The narrow range of the limits of agreement suggests that the model's performance is consistent across the range of earthquake intensities.

**Figure 7: Qualitative results of PaperOwl for Multimodal Diagram Analysis with outline or not. Wrong descriptions are marked as red. Descriptions that conform to diagrams but are inconsistent with the ground truth are marked as blue.**

**Table 7: The ablation study about the training strategy for Multimodal Diagram Captioning and Analysis performance. 'VA' means whether to fine-tune the Vision Abstractor. 'Crop' means whether to use the Croping Module.**

| | Setting | | Captioning | | Analysis | | | |
|---|---|---|---|---|---|---|---|---|
| | Img | VA | Crop | M | C | M | C | $F1^{gpt}$ | $C^{gpt}$ |
| r1 | 224 | ✗ | ✗ | 5.94 | 23.73 | 16.70 | 18.73 | 0.29 | 8.78 |
| r2 | 224 | ✓ | ✗ | 6.89 | 22.18 | **17.49** | 21.85 | **0.41** | 11.23 |
| r3 | 448 | ✓ | ✗ | 6.83 | 21.86 | 17.45 | 22.94 | 0.40 | 11.46 |
| r4 | 448 | ✓ | ✓ | **7.19** | **25.50** | 17.38 | **22.98** | 0.40 | **11.62** |

## 5.3 Qualitative Results

Fig. 7 presents qualitative results of *Multimodal Diagram Analysis* with outline or not. With preceding texts as the input and a simple [*Outline*] as the control signal, PaperOwl generates a paragraph analysis following the [*Outline*] and describes more details about diagrams. However, PaperOwl still makes some mistakes about the concrete numbers in the figure, showing the challenge of accurately understanding details among multiple scientific diagrams. Without the [*Outline*], PaperOwl could generate analysis related

to diagrams but different from the author's intention, showing the necessity of utilizing [*Outline*] as the control signal. More qualitative results of *Multimodal Diagram Captioning* can be found in the supplementary material.

## 6 CONCLUSION

Torwards a multimodal paper-writing copilot, we focus on enhancing the scientific diagram analysis ability of Multimodal LLMs. We first carefully build a multimodal dataset M-Paper based on high-quality Latex files of papers by aligning diagrams with captions and paragraph analysis. Simulating real scenarios of paper writing, we design Multimodal Diagam Captioning, Multimodal Diagram Analysis, and Outline Recommendation tasks. To better evaluate the analysis quality, we propose a GPT-based metric to measure both detailed n-gram matching and overall semantic similarity. We benchmark multiple state-of-the-art MLLMs and propose a strong baseline, PaperOwl, by performing instruction tuning on ensembled training data. Comprehensive experiments validate the effectiveness of incorporating preceding texts and outlines as inputs. Finally, our ablation study provides insights into model improvement, such as increasing image resolution to see more details and better balancing the multimodal information of context, outline, and diagrams.

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
