# OpenReview forum: "mPLUG-PaperOwl: Scientific Diagram Analysis with the Multimodal Large Language Model"
_acmmm.org/ACMMM/2024/Conference — MM2024 Poster_

### Official Review · Reviewer_a6fC · 2024-05-23

**Rating:** 5
**Confidence:** 4

**Summary:**

The mPLUG-PaperOwl paper presents a new dataset called M-Paper to enhance the diagram analysis capabilities of multimodal large language models (MLLMs) for assisting with academic paper writing. The M-Paper dataset includes scientific diagrams such as figures and tables, aligned with related text paragraphs, to support joint comprehension of multiple diagrams. The paper shows that training MLLMs on the M-Paper dataset leads to stronger performance on tasks like diagram captioning, analysis, and outline recommendation, compared to existing MLLMs. The key innovations are the multi-modal diagram understanding dataset, alignment of diagrams with text context, and the use of user-specified outlines to better align the model's capabilities with the writer's intent. The dataset, code, and models are proposed to be made publicly available to enable more versatile academic paper writing assistants.

**Strengths:**

* M-paper dataset is meaningful and useful for multi-modal research in the future.
* The paper is clear, easy to read, and well-motivated.
* Solid experiments and sota results.

**Limitations:**

Questions
* Why are equations longer than 40 characters dropped from the dataset ?
* Given that the M-Paper dataset consists primarily of computer science papers, how well would the model trained on this data generalize to papers from other domains like physics and math when performing *few-shot* or *zero-shot* learning?

**Suitability:**

3

---

### Official Review · Reviewer_JFGV · 2024-05-25

**Rating:** 4
**Confidence:** 3

**Summary:**

This paper proposed a novel approach to enhancing the diagram analysis capabilities of Multimodal Large Language Models (MLLMs), aimed specifically at assisting in academic paper writing. The authors introduced a new dataset, M-Paper, and present a comprehensive suite of tasks and metrics to benchmark the diagram comprehension abilities of the model. Overall, the proposed approach in this paper is interesting and effective for enhancing the multimodal comprehension capabilities of LLMs for scientific diagram analysis. The construction of the M-Paper dataset and the task design are also useful contributions to this field.

**Strengths:**

1. This paper focuses on an interesting and valuable to be investigated problem, the using scenarios are worth exploring.
2. The paper clearly defines three main tasks—Multimodal Diagram Captioning, Multimodal Diagram Analysis, and Outline Recommendation. The introduction of the GPT-based metric CIDErgpt for evaluating diagram analysis quality is a thoughtful approach that combines n-gram and semantic similarity, offering a betterassessment of model performance.
3. The authors conducted extensive experiments to validate the effectiveness of their approach. The ablation studies and comparisons with state-of-the-art MLLMs are thorough and provide valuable insights.

**Limitations:**

1. While the model shows strong performance in many areas, the qualitative results indicate that it sometimes struggles with accurately understanding detailed numerical information in diagrams. Could the authors explain that?
2. Conducting user studies to validate the effectiveness of the Outline Recommendation and the overall user experience with the paper-writing copilot would provide practical insights and further substantiate the model's real-world applicability.

**Suitability:**

2

---

### Official Review · Reviewer_wDyp · 2024-06-06

**Rating:** 5
**Confidence:** 3

**Summary:**

- This paper focuses on enhancing the scientific diagram analysis abilities of MLLMs to develop a more versatile copilot for academic paper writing.
- The authors build a new multi-modal dataset called M-Paper by parsing LaTeX source files of academic papers. M-Paper aligns diagrams in papers with related paragraphs to construct professional diagram analysis samples for training and evaluation. It is the first dataset to support joint comprehension of multiple scientific diagrams including figures and tables in image or LaTeX formats.
- The authors design three multimodal tasks simulating real paper-writing scenarios: Multimodal Diagram Captioning, Multimodal Diagram Analysis, and Outline Recommendation. A new GPT-based metric CIDErgpt is proposed to measure both n-gram and semantic similarity of the generated diagram analysis.
- Based on the DocOwl MLLM, the authors perform instruction-tuning on the combined training data from the three tasks to develop a strong baseline model called PaperOwl. Experiments show PaperOwl achieves stronger scientific diagram understanding performance.

**Strengths:**

- The authors introduce the CIDErgpt score, which combines the traditional CIDEr metric with a GPT-based semantic similarity evaluation and measures model performance more holistically.
- The paper conducts comprehensive ablation studies that provide valuable insights for further improving MLLMs on scientific diagram analysis, such as incorporating context and outline inputs, using higher image resolutions, and better multimodal information balancing.
- The paper is well-structured and clearly written.

**Limitations:**

- The effectiveness of the model and the proposed dataset are contingent upon the quality of the Latex sources and the accuracy of the diagram-caption alignments. In real-world scenarios, particularly with older papers or those not formatted in Latex, the utility of the model might be reduced. There's also the concern that inaccuracies in the initial parsing or diagram extraction could propagate errors through to the model's outputs.
- The paper aims to develop a copilot for academic paper writing. However, the evaluation is limited to automated metrics like CIDErgpt. Conducting a user study with researchers or students to assess the usefulness and effectiveness of the PaperOwl model in real-world paper writing scenarios would provide a more comprehensive evaluation.
- A more detailed analysis of the failure cases of the PaperOwl model would be informative. Understanding the common errors, such as misinterpreting diagram elements or generating irrelevant analysis, could guide future work on improving the model's robustness and reliability.

**Suitability:**

3

---

### Official Review · Reviewer_7Czs · 2024-06-07

**Rating:** 1
**Confidence:** 4

**Summary:**

The authors focus on enhancing the multi-modal diagram analysis capabilities of Multimodal LLMs. They developed a dataset called M-Paper by parsing LaTeX source files of high-quality papers and aligning diagrams with related paragraphs to create professional diagram analysis samples for training and evaluation. The claims that the paper is the first dataset to support joint comprehension of multiple scientific diagrams

**Strengths:**

1. The authors present a multimodal paper analysis dataset, and the task format is to generate an analysis of a paper containing text and figures inputs.

**Limitations:**

1. The writing of this paper is not clear,  e.g., the statistics of average input modalities and lengths are not present the statistics.

2. The paper does not present the quality of the targeted Diagram analysis of a sample. The caption of a table or figure in the paper may not be an analysis of the content.

3. They claim that the proposed dataset is the first one. Yet, they only change the input type compared to previous datasets and do not provide some new inputs or findings in the multimodal summarization.

4. The learning method and model are consistent with the existing open-source general multimodal large model. They do not present some structural changes to improve the performance on this task that needs long multimodal context inputs.

5. There are too few models for comparison. There are many large multimodal models in the community, but they only compared a few of them, e.g., gpt-4, gemini-pro.  Obviously, they did not meet the ACM MM standard to provide sufficient experimental evidence to support their analysis.

6. The cases are not clear and the font is so small.

Overall, I think this paper needs major revisions before it can be accepted.

**Suitability:**

2

---

### Meta-Review · Area_Chair_BWHW · 2024-07-03

**Recommendation:** Accept (Poster)
**Confidence:** 4

**Metareview:**

This paper addresses the task of scientific diagram analysis using Multimodal Large Language Models. A new multimodal diagram understanding dataset, M-Paper, is proposed by parsing Latex source files of high-quality papers.

This paper ended up with three positive reviews (Weak Accepts) and a negative one (Weak Reject).

Among its strengths, reviewers highlight:
- The problem addressed is relevant and the considered scenarios make sense (wDyp, JFGV, a6fC).
- Extensive experimental evaluation (wDyp,  JFGV, a6fC)
- The proposed dataset is relevant and valuable for the community (wDyp, JFGV, a6fC).
- A sound metric is proposed, combining CiDER and GPT-based semantic similarity (wDyp, JFGV).

Among its weaknesses, reviewers point out:
- Missing more in-depth discussion regarding the quality of the samples included in the proposed dataset and the impact that might have (7Czs, wDyp)
- A lack of a user study to complement automatic evaluation (wDyp, JFGV). Addressed in the rebuttal, according to reviewers wDyp and JFGV final responses.
- Limited number of compared multimodal baselines (7Czs)

To the best of my understanding, and as stated by reviewer 7Czs, some of his concerns were addressed. By considering the agreement between the other three reviewers on multiple positive aspects of the paper, I suggest it be accepted for publication.
However, clarifications provided by the authors in the rebuttal should be accommodated in the final version of the manuscript.